# Automatic foot ulcer segmentation using conditional generative adversarial network (AFSegGAN): A wound management system

**Jishnu P.**[1], **Shreyamsha Kumar B. K.**[1], **Srinivasan Jayaraman**[2]*

**1** TCS Research, Digital Medicine and Medical Technology- B&T Group, TATA Consultancy Services, Bangalore, Karnataka, India, **2** TCS Research, Digital Medicine and Medical Technology- B&T Group, TATA Consultancy Services, Cincinnati, Ohio, United States of America

* srinivasa.j@tcs.com

## Abstract

Effective wound care is essential to prevent further complications, promote healing, and reduce the risk of infection and other health issues. Chronic wounds, particularly in older adults, patients with disabilities, and those with pressure, venous, or diabetic foot ulcers, cause significant morbidity and mortality. Due to the positive trend in the number of individuals with chronic wounds, particularly among the growing elderly and diabetes populations, it is imperative to develop novel technologies and practices for the best practice clinical management of chronic wounds to minimize the potential health and economic burdens on society. As wound care is managed in hospitals and community care, it is crucial to have quantitative metrics like wound boundary and morphological features. The traditional visual inspection technique is purely subjective and error-prone, and digitization provides an appealing alternative. Various deep-learning models have earned confidence; however, their accuracy primarily relies on the image quality, the dataset size to learn the features, and experts' annotation. This work aims to develop a wound management system that automates wound segmentation using a conditional generative adversarial network (cGAN) and estimate the wound morphological parameters. AFSegGAN was developed and validated on the MICCAI 2021-foot ulcer segmentation dataset. In addition, we use adversarial loss and patch-level comparison at the discriminator network to improve the segmentation performance and balance the GAN network training. Our model outperformed state-of-the-art methods with a Dice score of 93.11% and IoU of 99.07%. The proposed wound management system demonstrates its abilities in wound segmentation and parameter estimation, thereby reducing healthcare workers' efforts to diagnose or manage wounds and facilitating remote healthcare.

## Author summary

Wound care protocol refers to setting systematic and evidence-based guidelines that healthcare providers follow to assess, manage, and treat wounds. Effective wound care is essential to prevent further complications, promote healing and reduce the risk of

**Data Availability Statement:** Open-source data provided by foot ulcer segmentation dataset of Diabetic Foot Ulcer Grand Challenge 2021:

Evaluation and Summary at https://arxiv.org/abs/2111.10376.

**Funding:** This research was supported by the Internal TCS Research funding. The funders had no role in study design, data collection and analysis, decision to publish, or preparation of the manuscript.

**Competing interests:** The authors have declared that no competing interests exist.

infection and other health issues. In addendum, accurate wound quantitative assessment is crucial in developing appropriate treatment plans as different types of wounds require a unique approach to healing. The post-COVID effect has accelerated the remote or m-health technology in the healthcare sector. Thus, we sought to assess the feasibility of wound health and provide a healing rate, which facilities continuous monitoring of the wound to physician and educates the patient. Thereby it elevates the patient's confidence in using the app without assistance. The proposed system demonstrates the feasibility of remote wound care for continuous monitoring.

## Introduction

The phenomenon of wound is called a 'silent epidemic' [1]. The victims of this often undergo physical, mental, and social consequences that are not recognized. The root causes may vary, like surgical intervention, injury, or other pathological conditions such as bed soar or pressure ulcer, diabetes, or vascular diseases. For example, foot ulcer, particularly in diabetes mellitus, plays a crucial role in the individual's lifestyle. Acute and chronic non-healing wounds significantly reduce those patients' quality of life [2].

Further, wounds require periodic examination and thorough treatment to prevent deterioration [3]. Ignorance can result in severe complications such as limb amputations and death [4, 5]. It has been estimated that the medical cost for acute and chronic wound treatment is around $96.8B in the United States. In other wound categories, it is estimated that pressure ulcers will be $4.5B by 2024 (2.5M people annually), Diabetes Foot Ulcer cost $9-$13B (estimated $64B by 2026), and Venous Ulcers $2.5B in the US alone [5]. In addendum, it has been reported that 80-85% of total treatment cost is nursing time and dressing charges. Eventually, the burden increases in the chronic wound segment, as they do not progress through the phases of healing in an orderly and prompt manner, requiring hospitalization and further treatment that adds billions in cost for health care services annually.

Wound care is considered cost-effective only when it is economical in the time and cost involved. To reduce the wound dressing cost and time, negative pressure wound therapy (NPWT) [6] has endeavored. However, NPWT or standard therapy largely depends on wound monitoring or wound characteristics such as width, length, area, and volume that need to be measured. In addition, the wound site or bed diagnosis is critical, which allows us to evaluate the effect of treatment or healing process [7]. Most healthcare professionals depend only on imprecise manual measurement and visual assessment of wounds [8], which is time-consuming and often inaccurate, hurting patients such as infection risks, inaccurate measurements, and discomfort [9]. A promising alternative to address this issue is to collect images of the area and apply computer-aided techniques to segment the ulcer. Industrial 4.0 has acclimated well to the digitization process to handle the pandemic challenge, revolutionizing remote care. Although digitization is convenient, the segmentation and evaluation of wound images are challenging due to the complexities involved in the wound-capturing processes, such as lighting conditions and time constraints in clinical laboratories [10].

In recent years, many research groups have proposed various approaches to tackle the task of wound digitization, and segmentation has been categorized into conventional [11–14] and deep learning (DL) approaches [15–20]. The first category, which focuses on combining image processing techniques with or without machine learning approaches, faces the following challenges [18]: **i)** Feature parameters depend on the user or expertise experience as they are manually curated parameters to get empirically hand-crafted features, and **ii)** the hand-crafted

features are affected by illumination, image resolution, and skin pigmentation, and **iii)** The features are not immune to severe pathologies and rare cases that are impractical from a clinical perspective.

The above challenges of conventional machine learning and image processing-based methods have been addressed by the DL-based methods combining feature extraction and decision-making. The superior performance of AlexNet [21] on ImageNet classification has created much traction among the research community, which led to the exploration of DL on semantic segmentation [22, 23] and medical image analysis [24, 25]. Wang et al., [26] used the vanilla fully convolutional neural network (FCN) architecture [23] to estimate the wound area by segmenting the wounds. Then, the estimated wound areas and the corresponding images are considered as time-series data to predict the wound healing progress using a Gaussian regression function model. Goyal et al., [27] employed the FCN-16 architecture to classify the pixels, whether it belongs to the wound or not, resulting in a wound segmentation. Liu et al., [28] study attempted to replace vanilla FCN decoder with a skip-layer concatenation up-sampled with bilinear interpolation and appended pixel-wise Softmax layer at the last layer to carry out the segmented image. Scebba et al., [19] proposed a detect-and-segment (DS) algorithm for producing wound segmentation maps with high generalization capabilities. Anisuzzaman et al., [20] presented an automated wound localizer based on the YOLOv3 Model followed by segmentation and classification. Pholberdee et al., [17] proposed a DL and data augmentation model to segment each wound tissue separately, and different machine learning models are used for detection and segmentation tasks. Chino et al., [29] used a convolutional neural network (CNN) and encoder/decoder deep neural network for automatic skin ulcer segmentation. Zahia et al., [30] proposed a CNN-based model for tissue segmentation of pressure injury wounds with the help of manual pre-processing steps. Lu et al., [31] proposed a color correction and CNN model with a two-step pre-processing pipeline to segment the overall wound without tissue segmentation. Chang et al., [32] proposed a superpixel-assisted, region-based automated pressure ulcer segmentation using U-Net, DeeplabV3, PsPNet, FPN, and Mask R-CNN with ResNet-101 encoder. As Goodfellow et al., [33] proposed a generative adversarial network (GAN) by training the two networks, generator and discriminator, alternatively such that the former creates the synthetic data (such as images, audio, or text) that resembles the real data and latter tries to differentiate between the synthetic and real data. The GANs became popular as they can create more realistic synthetic data that resembles a given dataset and can be used wherever data are scarce for training the DL networks.

The success of GAN has encouraged researchers to explore it in medical image analysis, such as automated muscle segmentation in computed tomography scanning [34], dose prediction in prostate cancer patients [35]. Xun et al., [36] explored the GAN in medical image segmentation and reported that the GAN extended variants significantly improved the accuracy of the medical image segmentation due to its good generating ability and the capability to capture the data distribution. The closest work in scope to our proposed work is from Sarp et al., [37], where they used a hybrid wound segmentation and tissue classification algorithm by exploiting the conditional generative adversarial network (cGAN) to learn directly from data without human knowledge.

In contrast, our proposed AFSegGAN network is based on cGAN for wound segmentation, which consists of U-Net for the generator network and CNN for Discriminator networks. In addition, we perform post-processing to determine the morphological features like wound width, length, area, and shape from the segmentation masks. Our contribution can be summarized as follows:

1. to build an Automatic Foot ulcer GAN (AFSegGAN) network with a scalable capability toward wound segmentation.

2. to estimate the morphological features using morphology and connected component operators, and to determine the closest wound shape that would help the wound dressing.

3. we developed a wound management system that facilitates mobile health care by integration of the AFSegGAN and Morphological Feature Estimation (MFE) modules.

4. AFSegGAN proposed here outperformed, specifically when the input image's quality is poor, where even manual annotation is difficult, or fails to detect the boundary manually.

5. as our system is developed as an API approach, it is easy to integrate with the Health care system.

## Methodology

### Wound management system (WMS) overview

The WMS based on AFSegGAN consists of 3 modules, 1) Patient portal, 2) Physician portal, and 3) AWS cloud platform, as shown in Fig 1. The patient portal is the mobile app that captures the wound as an image using the mobile camera. The captured wound images are uploaded to the cloud along with the meta information like patient name, patient id, wound location, time, and date stamp information.

The Physician portal is central to the WMS, showing the patient details, wound images over the event period, and the analyzed wound images with morphological metrics and shape information overlaid on the wound images. Here, the wound images are segmented using the AFSegGAN API module, and the morphological metrics and wound shape information are estimated using MFE API module. Here, the Physician access patient's records using the Patient ID, where the newly uploaded wound images of the patient are available for analysis and review. The morphological metrics, such as area, perimeter, width and length, and diameter, are also displayed in tabular form for analysis by the Physician. Further, the current and previous medications prescribed by the Physician are available along with the wound history plot of the morphological metrics over time. Remarks or prescriptions made by the Physician and the calculated wound healing metrics are stored in the cloud database. Additionally, the analyzed images get uploaded to the cloud database for future review. All the wound images,

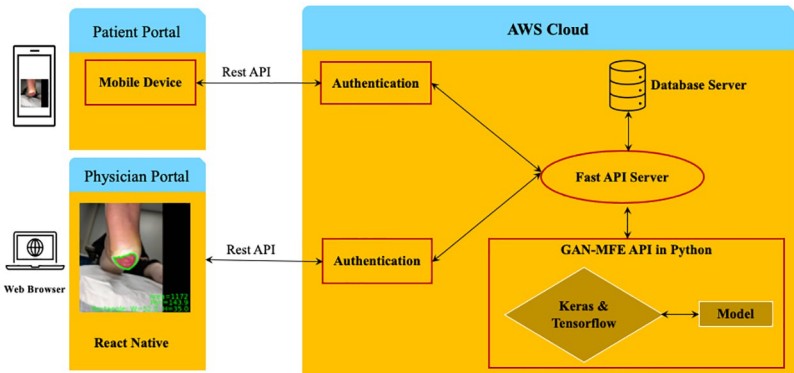

**Fig 1. Wound management system architecture.**

patient information, and API's of AFSegGAN and MF modules are stored in the cloud platform hosted by the health care provider/hospitals.

## AFSegGAN's architecutre

In this work, we propose the conditional generative adversarial network (cGAN) based wound segmentation and estimate the wound parameters. The main framework of the proposed approach (AFSegGAN) consists of the training and testing phases, as shown in Fig 2. The proposed system consists of two-step validation a) GAN module validation for wound segmentation and b) wound parameter estimation based on the segmented wound.

Generative modeling is an unsupervised machine learning task that involves discovering and learning the regularities or patterns in input data in such a way that the model can be used to generate output from new examples that plausibly could have been drawn from the original dataset. The proposed GAN architecture model is shown in Fig 3. As shown in Fig 3, the generator network is symmetric and consists of the U-Net architecture. The generator network contains two major parts: a) the encoder follows a general convolutional process and collects the context information at the bottleneck, and b) the decoder consists of the transposed 2D convolution layers that will generate the mask output from the context stored in the bottleneck. Skip connections are utilized to improve the quality of the generator-generated image by linking the corresponding encoder and decoder layers.

The discriminator network is different from a regular GAN discriminator as it classifies each patch of the input image separately rather than the entire image. There are six stacks of convolutional layers; each stack includes BatchNormalization layers and the LeakyRelu activation layers except in the last stack, where the Sigmoid activation layer is used, as shown in Fig 3. The discriminator model takes input from the source domain (original image ($x$)) and an image from the target domain (target mask ($y$) or generated mask $G(x)$) and predicts the likelihood of whether the image from target domain is a real or fake (generated version of the

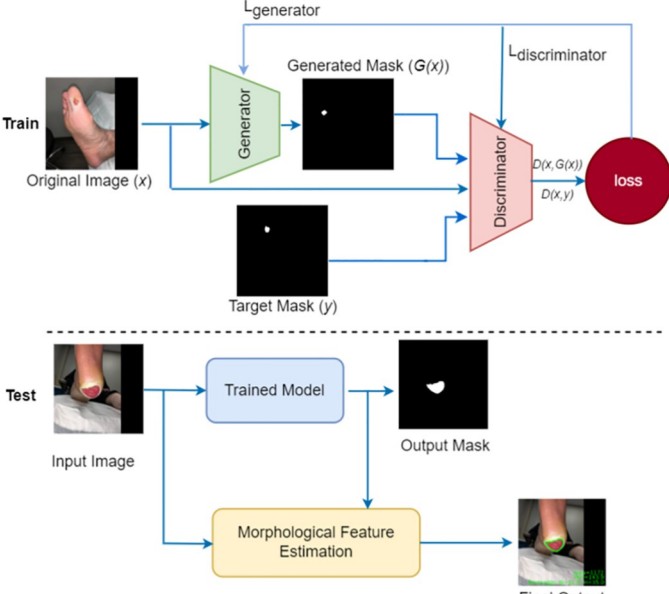

**Fig 2. AFSegGAN overview for wound segmentation.**

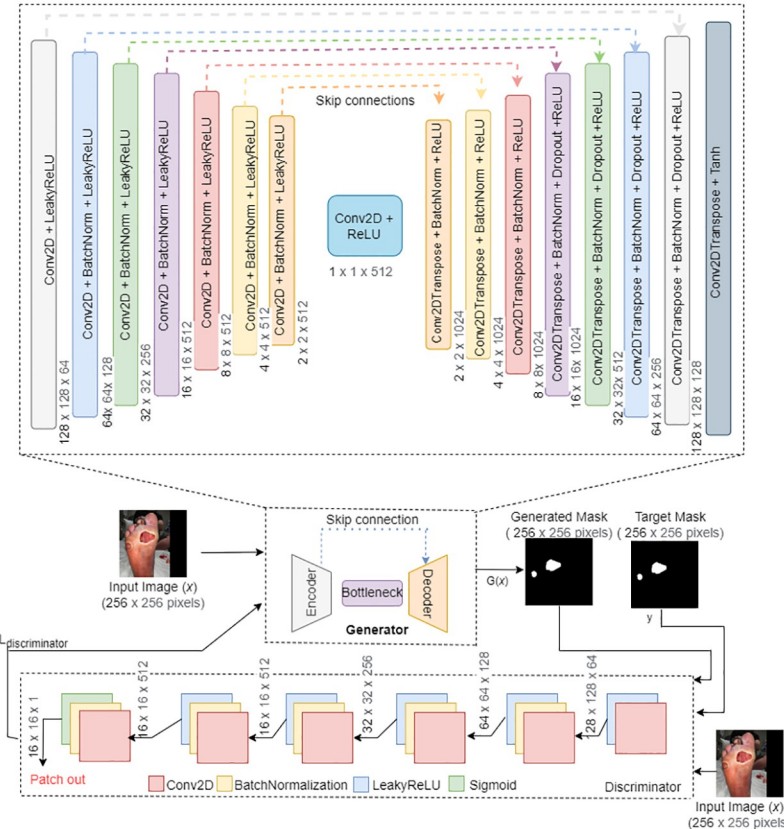

**Fig 3. Proposed AFSegGAN model architecture.**

original image). The discriminator network produces a matrix of values where each element corresponds to the respective patch of the input image. Patch-out is a single scalar value computed from the matrix's average. The generator network will be able to produce high-quality images due to this fine-grained feedback from the discriminator.

**AFSegGAN module training.** The GAN network training process is tricky as the sub-models (Generator and discriminator) compete, leading to network divergence. The nature of the optimization problem being solved will change whenever one of the model's parameters is updated. This will lead to training issues like mode collapse, vanishing gradient, non-convergence, and over-fitting, which can be overcome by balancing the generator and discriminator training. The discriminator and generator losses, as defined in Eqs 1 and 3, are used to address the above issues and to balance the training. Adversarial loss with a scaling factor $\alpha$, as represented in Eq 2, is used for training the discriminator

$$L_{discriminator} = \alpha \odot L_{adversarial}(G, D), \alpha = 0.5 \qquad (1)$$

$$L_{adversarial}(G, D) = \mathbb{E}[logD(x, y)] + \mathbb{E}[log(1 - D(x, G(x)))] \qquad (2)$$

Where, $G$—Generator, $D$—Discriminator, $x$—input image, $y$—target mask, $\mathbb{E}$ represents the expected value of the outputs of the discriminator and the Generator over the real and generated samples, $G(x)$ is the Generator's output for the given input image $x$, and $D(x, y)$ is the discriminator's output for the given input pair $(x,y)$. $D(x, G(x))$ is the discriminator's output

given a fake image $G(x)$ generated by the generator and the input image $x$. In order to reduce the impact of the discriminator loss on the generator training, compound loss, as represented in Eq 3, is used for training the Generator.

$$L_{generator} = \lambda_1 \odot L_{adversarial}(G, D) + \lambda_2 \odot L_{reconstruction} \qquad (3)$$

where $\lambda_1 = 1$, and $\lambda_2 = 100$ and $L_{reconstruction}$ is the loss function calculated by the pixel level comparison of both the generated mask $G(x)$ and the target mask $y$.

$$L_{reconstruction} = \sum_{i=1}^{n} |y_i - G(x)_i| \qquad (4)$$

Where, $n$ = Total number of pixels, $y_i$—the observed value for the $i^{th}$ pixel and $G(x)_i$—the predicted value for the $i^{th}$ pixel. The training of the proposed AFSegGAN model for wound segmentation is summarized in Algorithm 1 for better understanding.

**Algorithm 1** Pseudo code of training AFSegGAN

```
Input: Training step = number of epochs × batch per epoch
Initialize the generator (G) and Discriminator(D) with random weights.
1. Sample a minibatch Xₙ of n samples x¹, ..., xⁿ from the set of Origi-
nal images x.
2. Sample a minibatch Yₙ of n samples y¹, ..., yⁿ from the set of target
masks y.
3. Generate a batch of fake Masks G(Xₙ) using the current generator
weights.
4. Train the discriminator on the real and fake batches separately:
  i Calculate the realLoss for the discriminator using the real batch
Xₙ & Yₙ.
  ii Calculate the fakeLoss for discriminator using the fake batch Xₙ &
G(Xₙ).
  iii Update the discriminator weights using the sum of the two losses
represented in Eq 1.
5. Train the Generator using the updated discriminator from step 4:
  i Calculate the generator loss for the batch of input images Xₙ using
the target masks Yₙ and the respective generator output (fake masks) G
(Xₙ) as mentioned in Eq 3.
  ii Update the generator weights using the generator loss.
6. Periodically save the generator weights to disk.
return: After training is complete, use the saved generator weights to
generate new data
```

## Morphological feature estimation (MFE) module

Morphological feature estimation was performed on the outcome of AFsegGAN generated mask as a post-processing step, as shown in Fig 4. The main intention of employing a morphological approach is to remove the small regions or spurious noises and to fill the tiny holes within the wound to improve the true positive rate. For instance, the deep learning network could identify the blood stain as a wound and increase the false positive rate. On the other hand, the abnormal tissue, like fibrinous tissue inside the wound, could be treated as a non-wound region by the network representing it as small holes inside the segmented mask. This study uses morphological operations for noise and hole removal and connected component analysis for labeling the wound regions. This contradicts Wang et al. [18] study, where the connected component analysis removes the noises and fills the tiny holes within the wound region. Also, Wang et al. [18] does not measure the wound dimensions. The connected component analysis used here is acclimated to label the region of interest (ROI) followed by

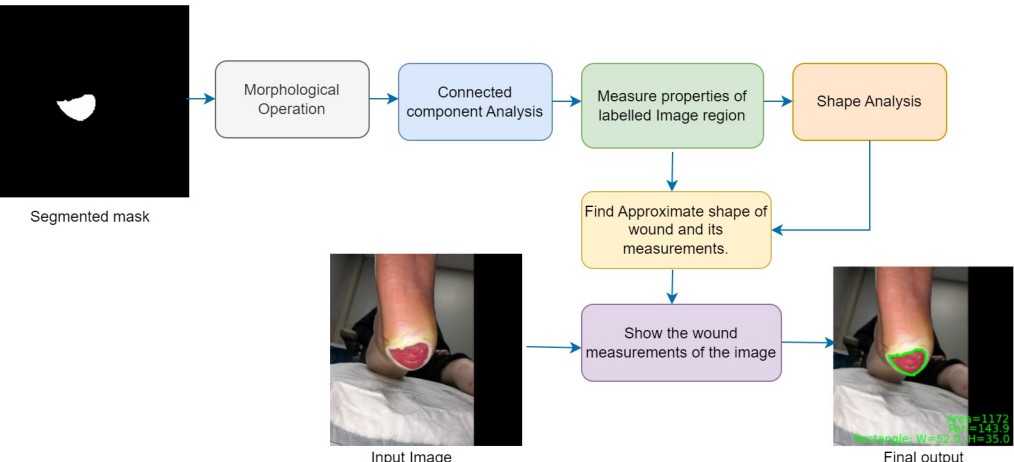

**Fig 4. Functional block diagram of morphological feature estimation module.**

measurement of those labeled connected areas. These labeled connected regions are used to estimate the wound dimensions such as width, length, circle diameter, major and minor axis length of the ellipse, area, and perimeter. These measurements are used in conjunction with the shape analysis algorithm to determine the approximate wound shape. To do this, the wound's eccentricity of the ellipse, the circularity, and the rectangularity are calculated using Eqs 5, 6 and 7, respectively, and then the approximate shape of the wound is determined.

$$e = \frac{foci}{length\ of\ major\ axis}, 0 \leq e \leq 1 \tag{5}$$

$$\Omega_c = \frac{A}{r^2}, \tag{6}$$

where $A$ = area and $r$ = perimeter

$$\psi_R = \frac{no\ of\ pixel\ in\ ROI}{no\ of\ pixel\ in\ boundary\ box} \tag{7}$$

## Results and discussion

The model is implemented in Python with Keras and TensorFlow as the backend. Training and validation are performed on HPC A3 TCS (TATA Consultancy Services) Server powered by NVIDIA DGX- A100 series.

### Dataset

In this study, we used the foot ulcer segmentation dataset [38], which is an extended version of the chronic wound dataset [18], and the same dataset is used as the training and testing sets in MICCAI 2021 foot ulcer segmentation challenge. The dataset contains 1010 images for training, 200 images for evaluation, and 278 images for testing. Further, the training dataset volume increased to 4040 by performing the augmentation techniques, including brightness improvement, saturation, rotation, and horizontal and vertical Flips. Before the augmentation process, all the images in the dataset are resized to 256 × 256 from 512 × 512 pixels.

## AFSegGAN module results

Adam optimizer with learning rate ($\eta$) = 0.002 and $\beta$ = 0.5 is used for updating the network parameters for both Generator and the discriminator. A batch size of 16 is used for training. In order to ensure stable training, at each training step, the best-performing model is saved based on loss value fluctuations. Here, *Training steps = number of epochs × batch per epoch*. The loss curve shown in Fig 5 indicates that the training started with high generator loss. Gradually, the Generator started learning the pattern and producing plausible outputs closer to the target mask.

AFSegGAN architecture performance was evaluated using the evaluation metrics like Dice Score, Intersection over Union (IoU), Precision, and recall, which are given below for completeness.

$$Precision = \frac{True\ Positive}{True\ Positive + False\ Positive} \tag{8}$$

$$Recall = \frac{True\ Positive}{True\ Positive + False\ Negative} \tag{9}$$

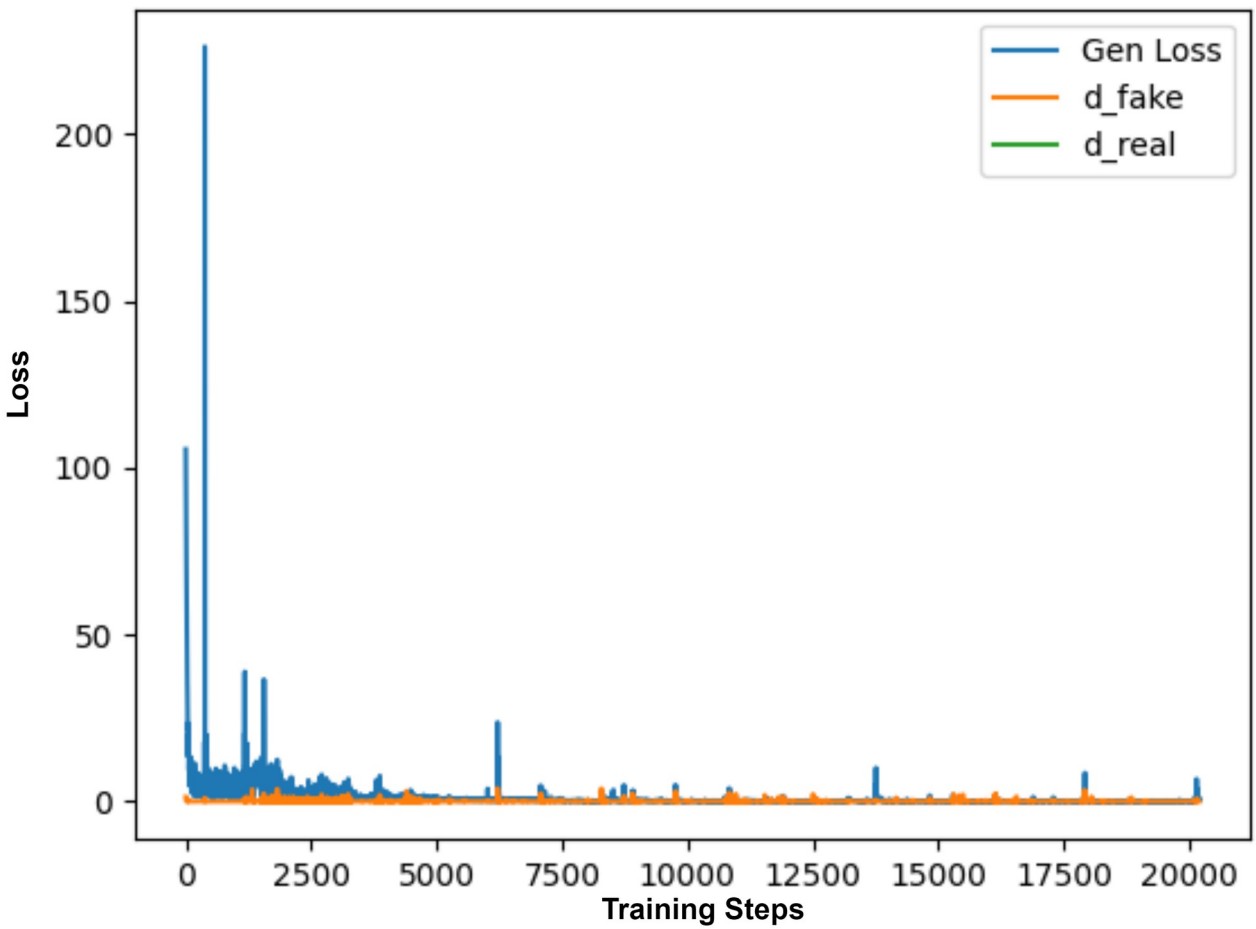

**Fig 5. AFSegGAN learning curve.** Training start with high fluctuations in generator loss and then acquires balance between Generator and discriminator as the training_steps increase.

$$Dice\ Score = \frac{2 \times True\ Positive}{2 \times True\ Positive + False\ Positive + False\ Negative} \tag{10}$$

$$IoU = \frac{True\ Positive}{True\ Positive + False\ Positive + False\ Negative} \tag{11}$$

We compared the segmentation performance of our model with the top 5 scorers in the MIC-CAI 2021 foot ulcer segmentation challenge and the simultaneous wound border segmentation and tissue classification using cGAN [37], as shown in Table 1. It is observed from Table 1 that the proposed method (AFSegGAN) has improved the dice score of Sarp et al. [37] from 90.00% to 93.11%, an increase of 3.45%. Note that Sarp et al. utilized both the original image and the latent space as the input to the Generator. However, our model uses only the original image for the Generator, and we applied new loss functions for both the discriminator and the Generator for training stability. The generator and discriminator networks are customized by capturing the diminutive details from the inputs to achieve better performance. Thus, our proposed model achieves a precision score of 94.04%, a recall score of 94.55%, a dice score of 93.11%, and an IoU score of 99.07%.

Further, the comparison between the proposed (AFSegGAN) and the other state-of-the-art methods, LinkNEt-EffB1, U-Net-EffB2, DeepLabV3+, and DeepLabV3+SE, indicates that the proposed model significantly performs better in terms of dice score by 1.1%, 1.3%, 1.3% and 0.9%, respectively, and 16%, 16.5%, 7.6%, and 7.2%, respectively, in terms of IoU score.

Similarly, AFSegGAN outperforms the other state-of-the-art methods, LinkNEt-EffB1, U-Net-EffB2, DeepLabV3+, and DeepLabV3+SE, by 3.5%, 3.2%, 7.9% and 7.1%, respectively, in terms of Recall. Also, AFSegGAN performs better than LinkNEt-EffB1 and U-Net-EffB2 methods by 1.2% and 1.9%, respectively, and performs poorer than DeepLabV3+ and DeepLabV3+SE by 2.4% and 2.0%, respectively, in terms of Precision. Considering the best performance of the proposed AFSegGAN in terms of Recall, Dice, and IoU scores than the other state-of-the-art methods, the segmentation mask obtained from the proposed method is used for Morphological Feature estimation.

**Table 1. Comparison of proposed model with other state-of-the-art models on MICCAI 2021 foot ulcer segmentation challenge dataset.**

|  | Precision [%] | Recall [%] | Dice [%] | IoU [%] |
|---|---|---|---|---|
| **AFSegGAN(Proposed Model)** | 94.04 | **94.55** | **93.11** | **99.07** |
| **Amirreza Mahbod et al., [39]** | 91.55 | 86.22 | 88.80 | - |
| **Zhang Yichen [39]** | 88.87 | 86.31 | 87.57 | - |
| **Bruno Oliveira [39]** | - | - | 87.06 | - |
| **Adrian Galdran [39]** | 90.03 | 84.00 | 86.91 | - |
| **Jianyuan Hong [39]** | - | - | 86.27 | - |
| **LinkNet-EffB1 [39]** | 92.88 | 91.33 | 92.09 | 85.35 |
| **U-Net-EffB2 [39]** | 92.23 | 91.57 | 91.90 | 85.01 |
| **Sarp et al., [37]** | - | - | 90 | - |
| **DeepLabV3+ [40]** | **96.4** | 87.6 | 91.9 | 92.4 |
| **DeepLabV3+SE [40]** | 96 | 88.3 | 92.3 | 92.4 |

## Morphological feature estimation (MFE) module results

In the post-processing step, the identified wound boundaries are laid over the original image, and the morphological features are estimated. Additionally, morphology operations are performed on the binary mask to remove the tiny regions/spurious noises and to fill the small holes within the wound.

Fig 6 shows the wound images, segmented masks obtained from the proposed model, and the respective post-processed outputs. Here, column (a) represents the input wound image, column (b) represents the segmented output, and column (c) represents the output after the morphological feature estimation. To be specific, it is observed that the morphological operator extracts the wound region precisely as represented with a green boundary as shown in Fig $6(c_1)$, $6(c_2)$ and $6(c_3)$ for the corresponding input images ($a_1$, $a_2$, $a_3$), respectively. Further, it determines the wound shape as an ellipse for Fig 6 ($c_1$), as we can correlate this to the optimized shape of the bandage used to cover this wound. In addendum, the wound's area, perimeter, and ellipse parameters, such as major-axis and minor lengths, are also estimated during the post-processing and are overlaid on the wound image (Fig $6(c_1)$). Further, as shown in the Fig $6(c_2)$, the wound shape is estimated as a rectangle; even though the wound ROI shape is not precisely the rectangle, the physician would recommend using a rectangular shape bandage or wound dressing. For a rectangular shape, the width and height are estimated and

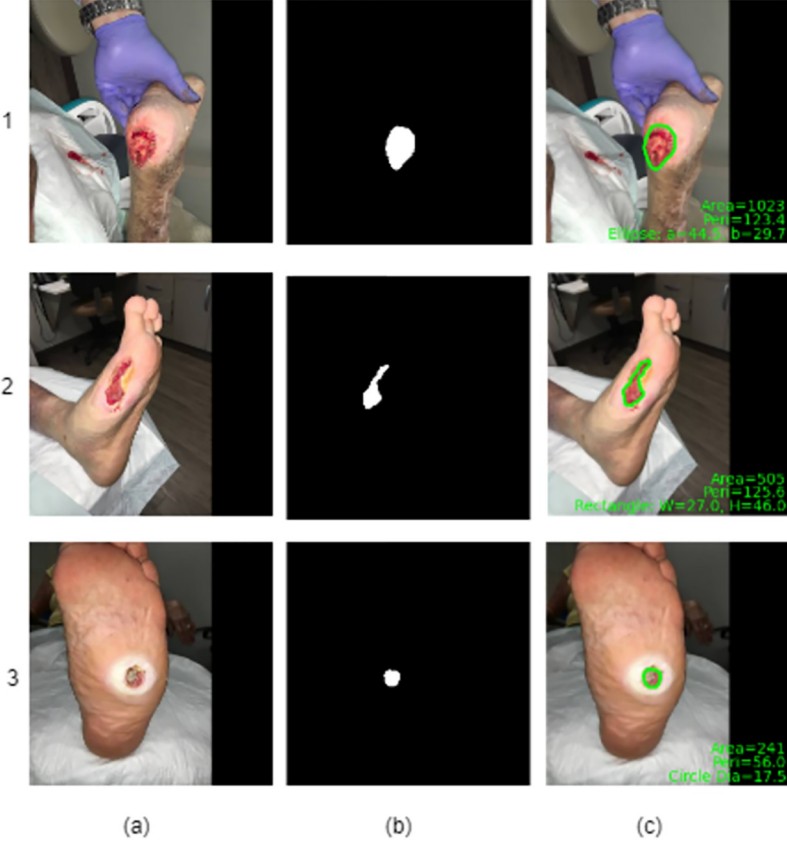

(a)                    (b)                    (c)

**Fig 6. MFE module results.** (a) shows the input image, (b) shows the generator output of the proposed model, (c) shows the result after the morphological metrics such as area, perimeter, and shape (rectangle, circle, and ellipse) with the Green color visual layout representing the wound boundary on the original wound image.

overlaid on the wound image, along with the area and perimeter. Similarly, in Fig 6($c_3$), as the wound shape is circular, the diameter, area, and perimeter are estimated and overlaid on the wound image.

## Impact of the proposed AFSegGAN

The proposed method performs well even for images with low contrast, low resolution, and more minor wounds, where the manual annotation by the expert was complex. In these scenarios, as the ground truth labels are not annotated, the dice score for that images will be near zero, thereby indicating that our model is under-performed, but it is not so. For example, Fig 7 shows such two exceptional cases where the ground truth of the wound is not annotated due to i) poor image quality (less contrast) and ii) multiple wounds with slim sizes.

These worst-case scenarios may affect the model performance as various factors affect the image-capturing process in clinical laboratories [13]. However, our model segmented those wound regions as shown in $c_1$, $c_2$, and finally estimated the morphological measures as shown in $d_1$ and $d_2$. Here, column (a) represents the input image, column (b) indicates the ground truth labels that do not have annotated or missing annotation, the GAN model output is shown in column (c), and the final output after morphological feature estimation in (d). To be more specific, Fig 7's column (d), where the multiple wound regions were marked with different colors like green, yellow, and blue, is a piece of clear evidence that our model surpasses compared to other techniques. In addition, our method also achieved good Precision and a competitive recall score compared to other methods. In yet another aspect, our model reduces the dependence on the large dataset to learn the wound features. To evaluate the performance of the proposed method with limited training data, we compared the proposed model with DeepLabV3+ and DeepLabV3+SE models in Table 1, which were trained on the largest dataset of 51362 images after augmentation [40], and observed that the proposed method outperforms both DeepLabV3+ and DeepLabV3+SE in terms of Recall, Dice, and IoU.

## Conclusion

Computer-aided foot ulcer segmentation techniques can be effectively used as an alternative to manual inspection and subsequent wound region quantification. In this paper, we proposed a

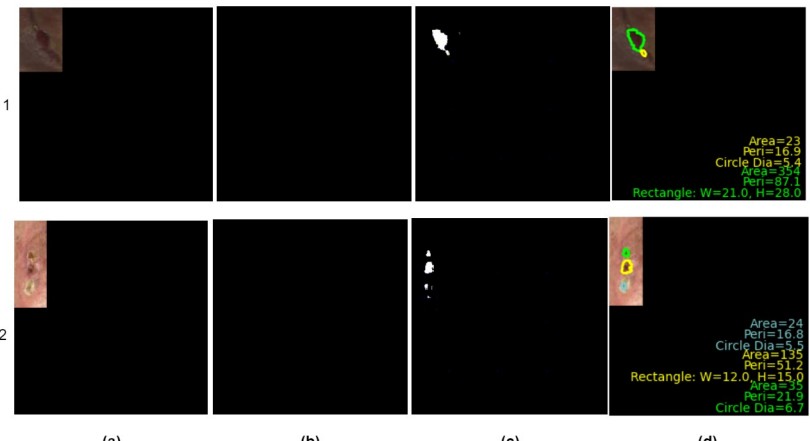

**Fig 7. Effectiveness of proposed method in handling low quality image.** (a) shows the input image, (b) shows the target mask, but in this case, it is not annotated due to image quality, (c) shows the mask generated by AFSegGAN and (d) results after morphological feature estimation. Green, yellow, and blue color visual layouts represent the first, second, and third wounds in the original wound image, respectively.

conditional generative adversarial network(cGAN) based segmentation model to detect the peripheral wound boundary from the foot ulcer clinical images, followed by a morphological feature estimation module to estimate the wound parameters. This study finding indicates that the proposed AFSegGAN model provides a high dice and outstanding IoU score compared with other state-of-the-art models. Thus, a developed wound management system that integrates the AFSegGAN model with accurate boundary detection and the wound parameters module, i.e., the MFE module, will assist the physician in early wound closure prediction and healing index assessment, empowering the pervasive wound care system.

## Acknowledgments

The authors acknowledge the HPC–TCS B&TS team for providing us with the ultra-modern HPC infrastructure, which accelerated and supported this work immensely.

## Author Contributions

**Conceptualization:** Srinivasan Jayaraman.

**Methodology:** Jishnu P., Shreyamsha Kumar B. K.

**Supervision:** Srinivasan Jayaraman.

**Writing – original draft:** Jishnu P., Shreyamsha Kumar B. K.

**Writing – review & editing:** Srinivasan Jayaraman.

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
