## [Decision Letter · Decision Letter 0]

7 Aug 2023

Automatic Foot Ulcer Segmentation using Conditional Generative Adversarial Network (AFSegGAN): A Wound Management System

PDIG-D-23-00192

Dear Dr. Jayaraman,

We are pleased to inform you that your manuscript 'Automatic Foot Ulcer Segmentation using Conditional Generative Adversarial Network (AFSegGAN): A Wound Management System' has been provisionally accepted for publication in PLOS Digital Health.

Best regards,

Ryan S McGinnis

Academic Editor

PLOS Digital Health

Thank you for your submission to PLOS Digital Health!

Reviewer Comments (if any, and for reference):

Reviewer's Responses to Questions

**Comments to the Author**

1. Does this manuscript meet PLOS Digital Health’s publication criteria? Is the manuscript technically sound, and do the data support the conclusions? The manuscript must describe methodologically and ethically rigorous research with conclusions that are appropriately drawn based on the data presented.

Reviewer #1: Yes

Reviewer #2: Yes

2. Has the statistical analysis been performed appropriately and rigorously?

Reviewer #1: I don't know

Reviewer #2: Yes

3. Have the authors made all data underlying the findings in their manuscript fully available (please refer to the Data Availability Statement at the start of the manuscript PDF file)?

Reviewer #1: Yes

Reviewer #2: Yes

4. Is the manuscript presented in an intelligible fashion and written in standard English?

Reviewer #1: Yes

Reviewer #2: Yes

5. Review Comments to the Author

Reviewer #1: The paper was written in a rigorous way.

The authors left no stone unturned, ensuring that even readers with limited prior knowledge in the field could understand. Each step was outlined clearly, allowing for easy replication and understanding. Such attention to detail is commendable and greatly enhances the paper's overall value.

The most compelling aspect of this paper, however, was the authors' demonstration that their proposed method outperformed other machine learning techniques in the context of wound management. By meticulously comparing their approach to existing methods, they provided substantial evidence of its superiority. This discovery is particularly exciting, as it offers promising prospects for improving the efficiency and effectiveness of wound management practices.

Reviewer #2: Thank you for the opportunity to review this manuscript. The approach presented here has the potential to advance mobile health delivery for wound care that may result in decreased complications. i.e. lower extremity amputations, and thereby reduce patient morbidity and mortality, as well as healthcare costs.

6. PLOS authors have the option to publish the peer review history of their article (what does this mean?). If published, this will include your full peer review and any attached files.

**Do you want your identity to be public for this peer review?** For information about this choice, including consent withdrawal, please see our Privacy Policy.

Reviewer #1: No

Reviewer #2: No
